# GG-GAN: A Geometric Graph Generative Adversarial Network

## Abstract

We study the fundamental problem of graph generation. Specifically, we treat graph generation from a geometric perspective by associating each node with a position in space and then connecting edges in-between based on a similarity function. We then provide new solutions to the key challenges that prevent the widespread application of this classical geometric interpretation: (1) modeling complex relations, (2) modeling isomorphic graphs consistently, and (3) fully exploiting the latent distribution. Our main contribution is dubbed as the geometric graph (GG) generative adversarial network (GAN), which is a Wasserstein GAN that addresses the above challenges. GG-GAN is permutation equivariant and easily scales to generate graphs of tens of thousands of nodes. GG-GAN also strikes a good trade-off between novelty and modeling the distribution statistics, being competitive or surpassing the state-of-the-art methods that are either slower or that are non-equivariant, or that exploit problem-specific knowledge.

## 1 Introduction

Learning distributions from empirical observations is a fundamental problem in machine learning and statistics (Goodfellow et al., 2014; 2016; Salakhutdinov, 2015; Foster, 2019). A challenging variant entails modeling distributions over graphs—discrete objects with possibly complex relational structure (Simonovsky & Komodakis, 2018; You et al., 2018; De Cao & Kipf, 2018; Liao et al., 2019; Niu et al., 2020; Yang et al., 2019). When successfully trained, *deep* graph generative models carry the potential to transform a wide range of application domains, for instance by finding novel chemical compounds for drug discovery (De Cao & Kipf, 2018), designing proteins that do not exist in nature (Huang et al., 2016), and automatically synthesizing circuits (Guo et al., 2019).

By and large, there are four properties that a graph generator $g$ should possess: (1) *Isomorphism consistency*: $g$ should assign isomorphic graphs the same probability—a property also referred to as permutation equivariance (Niu et al., 2020; Yang et al., 2019). (2) *Expressive power*: $g$ should be able to model local and global dependencies between nodes and graph edges, e.g., going beyond simple degree statistics and learning structural features and motifs. (3) *Scalability*: $g$ should be able to synthesize graphs with tens of thousands of vertices. (4) *Novelty*: $g$ should produce non-isomorphic graphs that are similar to (but not necessarily *in*) the training set.

Property (1) is important since there exist exponentially many ways to represent the same graph as a vector, inconsistent methods effectively waste a large portion of their capacity in describing different ways to construct the same object. Properties (2) and (3) are critical in large-scale contemporary applications that require going beyond simple degree statistics as well as learning structural features and motifs towards simultaneously modeling *local* and *global* dependencies between nodes and graph edges (You et al., 2018). Property (4) is natural since common failure modes for graph generators include memorizing the training set and repeatedly generating the same graphs.

### 1.1 Geometric graph generation

Aiming to satisfy these properties, we propose a geometric generator that represents graphs spatially by embedding each node in a high-dimensional metric space and then by connecting two nodes if their positions are sufficiently similar. There is precedence to our approach, as spatial representations of graphs have been heavily used to construct simple models of random graphs, such as random geometric graphs, unit-disc graphs, unit-distance graphs, and graphons (Huson & Sen, 1995; Penrose et al., 2003; Bollobás et al., 2007; Lovász, 2012; Alon & Kupavskii, 2014; Glasscock, 2015).

Surprisingly, these classical geometric approaches have so far found limited adoption in the context of *deep* generative graph models. In fact, it is easy to verify empirically that a direct, naive application of these methods yields poor performance, requiring additional conditioning and other stabilizing procedures to train effectively (Yang et al., 2019; Serviansky et al., 2020).

Our work precisely bridges this gap, showing how deep geometric graph generation approaches can perform well. Our contributions are two-fold:

*I. We shed light into the fundamental limits and challenges of geometric graph generators (Section 2).* We derive sufficient conditions for representing graphs spatially and demonstrate that for sparse graphs the embedding dimension can depend only logarithmically on $n$. We then identify challenges that arise when building powerful and isomorphism-consistent generators. Interestingly, we find that straightforward generators must solve a non-trivial collision avoidance problem for *every* graph. We present evidence that such generators cannot be easily trained even in simplified supervised settings.

*II. We avoid collisions at generation time, while retaining scalability and consistency (Section 3).* We propose geometric graph generative adversarial networks (GG-GAN) with new twists. Our numerical evidence demonstrate that our proposed changes can have considerable impact on graph generation quality, and can capture complex relationships. Furthermore, GG-GAN is significantly faster that SotA (autoregressive) models, while also being competitive in terms of captured statistics and novelty. A case in point, our method can generate graphs of 10k nodes in $\sim 0.2$ seconds, which is 2 orders of magnitude faster than the fastest autoregressive method within the state of the art.

## 1.2 RELATION TO EXISTING WORK

We argue that despite the impressive progress so far, no current approach satisfactorily meets the four aforementioned properties. Isomorphism inconsistent methods (Kipf & Welling, 2016; You et al., 2018; Bojchevski et al., 2018; De Cao & Kipf, 2018; Liao et al., 2019) tend to memorize the training set in the absence of problem-specific rewards (De Cao & Kipf, 2018). Autoregressive variants, in particular, possess large expressive power, but are limited in scalability as they construct the graph one node or block at a time (Bojchevski et al., 2018; Li et al., 2018; You et al., 2018; Yang et al., 2019; Liao et al., 2019). Clever optimization and exploitation of sparsity can help (Dai et al., 2020), but only to a degree, as can be seen in our experiments.

The closest methods to ours are ScoreMatch (Niu et al., 2020), Set2Graph (Serviansky et al., 2020) and CondGen (Yang et al., 2019), all of which are consistent and non-autoregressive. ScoreMatch samples graphs from a learned score function via annealed Langevin dynamics, an approach that we find works well only for small graphs. CondGen combines a variational autoencoder (VAE) with a conditional GAN for improved training stability. NetGAN (Bojchevski et al., 2018), while being a pure GAN does not operate on distributions of graphs. Instead, it learns to generate random walks on a single large graph using an autoregressive generator and then assembles the graph from these random walks. TagGen (Zhou et al., 2020) uses a similar model on temporal interaction graphs, but instead of learning a generator, it trains a transformer critic to score random temporal walks and then samples from them in parallel via rejection sampling and assembling the final graph. Neither model directly captures a graph distribution, but only addresses the distributions of the random walks.

Our work alleviates the need for such modifications or indirect modeling, demonstrating that a pure GAN approach suffices when the generator has been set-up appropriately. Set2Graph also adopts a geometric perspective to solve partitioning, Delaunay triangulation, and convex hull problems and not to generate graphs. By identifying and solving the collision avoidance problem, our work improves the efficacy of Set2Graph-type approaches, such as ours, in implicit deep generative modeling.

**Notation.** In the sequel, upper-case letters refer to sets, and bold-face letters denote vectors and matrices. We represent any set $X = \{\boldsymbol{x}_1, \ldots, \boldsymbol{x}_n\}$ of $n$ points as a matrix $\boldsymbol{X}$ with each row corresponding to some point $\boldsymbol{x}_i$ in arbitrary order. Functions, such as neural networks, are denoted by lower case letters, such as $f, g$. Every undirected weighted graph $G = (V, E, w)$ can be defined in terms of a set of $n = |V|$ nodes, a set of edges $E$ with $(v_i, v_j) \in E$ if there exists an edge joining the $i$-th and $j$-th nodes, and a weight function $w : E \to \mathbb{R}_+$ indicating the strength of each connection. A simple graph $G = (V, E)$ is an undirected graph without weights. Finally, the symbol $\mathfrak{S}_n$ refers to the group of permutations on $[n] = (1, \ldots, n)$. All proofs are presented in the appendix.

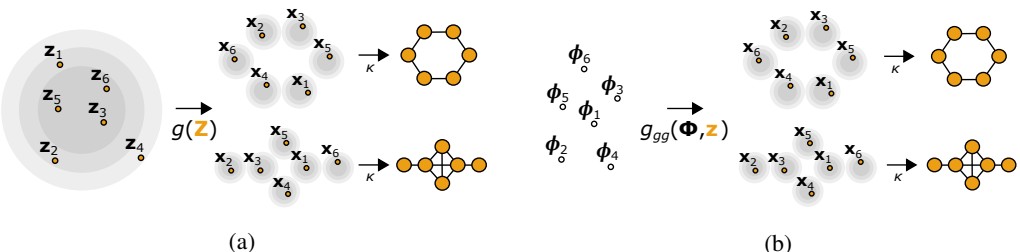

Figure 1: (a) Methods that attempt to learn a mapping between a random input $\boldsymbol{Z} = [\boldsymbol{z}_1, \ldots, \boldsymbol{z}_n]^\top$ and a target configuration $\boldsymbol{X} = [\boldsymbol{x}_1, \ldots, \boldsymbol{x}_n]^\top$ need to ensure that no two input points are mapped to the same output separately for *each* draw of $\boldsymbol{Z}$. Learning to perform collision avoidance is non-trivial for permutation equivariant $g$. (b) We instead learn a distribution over functions $g_{gg}(\cdot, \boldsymbol{z})$ operating on a fixed point set $\Phi$. Our method can learn to assign a different role to each point $\boldsymbol{\phi}_i$ during training, side-stepping the need for collision avoidance at every random draw of $\boldsymbol{z}$.

## 2 FUNDAMENTALS OF GEOMETRIC GRAPH GENERATION

This section recalls some basics of geometric graph generation, presents sufficient conditions for representing graphs spatially (Section 2.1), and identifies key challenges (Section 2.2) preventing their wide-spread application. Our observations motivate GG-GAN—presented in Section 3.

### 2.1 REPRESENTING GRAPHS SPATIALLY

We can represent graphs simply by placing $n$ nodes in some metric space and then by connecting them if their position satisfies some notion of similarity (Huson & Sen, 1995; Penrose et al., 2003; Bollobás et al., 2007; Lovász, 2012; Alon & Kupavskii, 2014; Glasscock, 2015).

More concretely, let $\boldsymbol{x}_1, \ldots, \boldsymbol{x}_n$ be $n$ points in $\mathbb{R}^k$. We determine whether graph $G = (V, E)$ of $n = |V|$ nodes contains edge $(v_i, v_j) \in E$ by testing if $\kappa(\boldsymbol{x}_i, \boldsymbol{x}_j) = 1$, for a similarity function $\kappa$.

It is not hard to see that this particular representation is complete[1] over the space of undirected graphs: every undirected graph can be represented by $n$ points as long as $k$ is sufficiently large. For instance, if $\boldsymbol{x}_i$ is chosen to be the row of the matrix square-root of the adjacency matrix $\boldsymbol{A}$ of the graph $G$, and $\kappa(\boldsymbol{x}_i, \boldsymbol{x}_j) = \boldsymbol{x}_i^\top \boldsymbol{x}_j$, then the representation is complete if and only if $k = n$.

Surprisingly, the embedding dimension $k$ can favorably depend on the sparsity of the graphs we wish to represent. Indeed, for simple graphs of degree at most $\Delta$, Maehara and Rödl showed that $k = 2\Delta$ dimensions suffice (Maehara & Rödl, 1990). This theorem unfortunately relies on a discrete and non-differentiable similarity function $\kappa$ and only holds for graphs whose edges have no weights.

As shown next, the aforementioned issues can be overcome by relaxing the node degree dependency:

**Theorem 1.** Let $G$ be a weighted undirected graph with adjacency matrix $\boldsymbol{A} \in [0,1]^{n \times n}$ and maximum (combinatorial) degree $\Delta$. For any $\epsilon \in (0, 1)$, there exist points $\boldsymbol{x}_1, \ldots, \boldsymbol{x}_n$ in $\mathbb{R}^k$ with $k = O\left(\frac{\Delta^2 \log n}{\epsilon^2}\right)$, for which $|\boldsymbol{A}(i, j) - (e^{\boldsymbol{x}_i^\top \boldsymbol{x}_j} - 1)| \leq \epsilon$ for all $i \neq j$.

Therefore, any *weighted* graph can be approximately represented by $n$ points in at most $k = O(\Delta^2 \log n / \epsilon^2)$ dimensions and this is possible using the differentiable function $\kappa(\boldsymbol{x}, \boldsymbol{y}) = e^{\boldsymbol{x}^\top \boldsymbol{y}} - 1$.

### 2.2 CHALLENGES OF GEOMETRIC GRAPH GENERATION

In the following, we expose three key challenges inherent to geometric generators.

---

[1] A vector representation of a set is *complete* if and only if there exists a surjective function between the representation vectors and the elements of the set.

### 2.2.1 THE CURSE OF INDEPENDENCE

The simplest way of constructing a generator is by sampling each point $x_i$ independently from a distribution $\mathcal{D}_x$. We refer to this approach as *random graph generation* (RGG) (Penrose et al., 2003). It is a direct corollary of our Theorem 1 that RGG can be used to generate any simple graph:

**Corollary 1.** Fix any weighted undirected graph $G = (V, E)$ of degree at most $\Delta$ and let $A$ be its adjacency matrix. For any $\epsilon \in (0, 1)$, there exists $\mathcal{D}_x$ supported on $\mathbb{R}^k$ with $k = O(\Delta^2 \log n / \epsilon^2)$ such that with strictly positive probability, $|A(i, j) - (e^{x_i^\top x_j} - 1)| \leq \epsilon$ for all $i \neq j$.

In a deep learning context, one may think of modeling $\mathcal{D}_x$ by passing a Normal distribution through a learned function, such as a multi-layer perceptron (MLP): $x_i = g(z_i)$, where $z_i \sim \mathcal{N}(\mathbf{0}, I_m)$, for every $i \in [n]$ and with $m$ being the dimension of the latent space. Unfortunately, as it is shown next, such generators cannot satisfactorily control the probability with which each set is generated.

**Proposition 1.** RGG generates any $\{x_1, \ldots, x_n\}$ with $x_i \neq x_j$ with probability at most $O(ne^{-n})$.

Random graph generators are therefore weak since they sample every point in the set *independently* of the other points: the larger $n$ is, the smaller the probability that some specific points are sampled.

### 2.2.2 MODELING ISOMORPHIC GRAPHS CONSISTENTLY

Naturally, we can introduce dependencies between points by expressing the joint distribution $\mathcal{D}_X$ of $X = [x_1, \ldots, x_n]^\top \in \mathbb{R}^{n \times k}$ as the push-forward measure of a random $Z \in \mathbb{R}^{n \times m}$

$$X = g(Z), \quad \text{with} \quad Z = [z_1, \ldots, z_n]^\top \text{ and } z_i \sim \mathcal{N}(\mathbf{0}, I_m),$$

or by generating each point sequentially conditioned on the previous by some autoregressive model (Li et al., 2018; You et al., 2018; Liao et al., 2019). In theory, these generators are capable of approximating any joint distribution $\mathcal{D}_X$. However, if not selected carefully, these models can waste an exponentially large portion of the generator's capacity in generating graphs that are just isomorphic.

To illustrate the problem, let us consider the case where $g$ is a surjective function, meaning that for every $X$ there exists $Z$ with $g(Z) = X$. Then, a large subset of its latent space can be devoted to expressing different permutations of the same points:

$$\forall X \in \mathbb{R}^{n \times k} \text{ and } \pi \in \mathfrak{S}_n, \quad \exists Z, Z' \in \mathbb{R}^{n \times k} \quad \text{such that} \quad X = g(Z) = \pi g(Z') = \pi X',$$

where $\mathfrak{S}_n$ is the group of permutations on $[n]$, with $|\mathfrak{S}_n| = n!$. In other words, $g$ can generally assign different probabilities to $X$ and $\pi X$, even if the corresponding graphs are simply isomorphic.

A more appropriate class of functions are those that are equivariant to permutation:

**Proposition 2.** If $g$ is permutation equivariant, then for every $X$ and $\pi \in \mathfrak{S}_n$, we have $\text{Prob}(X \sim \mathcal{D}_X) = \text{Prob}(\pi X \sim \mathcal{D}_X)$.

The above observation is consistent with the findings of Yang et al. (2019). Serviansky et al. (2020) also provides guidelines on how to construct such a function $g$.

### 2.2.3 AVOIDING COLLISIONS

While we seek an equivariant function that maps a random $Z$ to $X$, learning such functions can prove challenging even in toy examples (*cf.,* Section 3.2). Intriguingly, we recognize that a principal difficulty has to do with ensuring that the points are mapped in an equivariant manner without collisions, i.e., with no two rows $z_i, z_j$ of $Z$ being mapped to the same output.

For intuition, let us simplify the setting by supposing that we wish to generate the same fixed $X$ for every random input $Z$. Function $g$ can achieve this task in two ways: (1) it can memorize the output by disregarding its input, or (2) it can learn to transform each row of $Z$ to one of $X$. Geometrically, the latter corresponds to moving $n$ points from a random initial position to a fixed final target. For instance, $g$ could select an assignment $t : [n] \to [n]$ and match every $z_i$ to a target position $x_{t(i)}$, with $t(i) \neq t(j)$ for each $i, j$. Even though the aforementioned procedure is only an example, we note that any successful $g$ that does not disregard $Z$ should implement some collision avoidance mechanism (implying in turn the existence of an assignment $t$)—otherwise, $g$ cannot ensure that two different input points $z_i$ and $z_j$ will not collide onto the same target position.

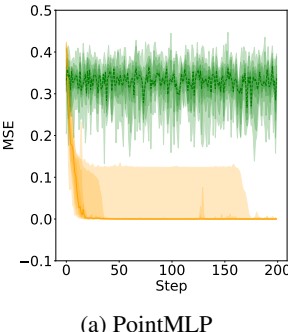 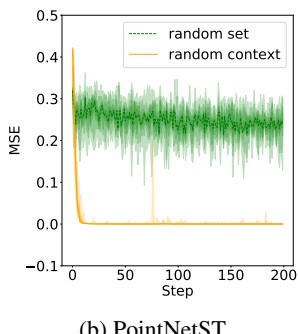 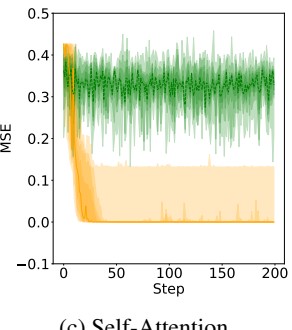

(a) PointMLP          (b) PointNetST          (c) Self-Attention

Figure 2: Mean-squared error per training step for three permutation equivariant networks tasked with generating a single target $\boldsymbol{X}$. Even with direct supervision, the networks do not learn when starting from randomly drawn points $\boldsymbol{Z}$ (*random set*–in green). The task is easily solved when starting from a fixed point set $\boldsymbol{\Phi}$ and a random vector $\boldsymbol{z}$ (*random context*–in orange).

Unfortunately, we argue that both these behaviors are unlikely to be learned in practice: In the first case, one has to learn to ignore any spurious relations that are induced by the random input. In the second case, due to being permutation equivariant, $g$ can rely on the position $\boldsymbol{z}_i, \boldsymbol{z}_j$ and *not* the order $i, j$ of the random points to tell them apart. *As such, the combinatorial collision avoidance problem needs to be solved anew every time that the points change, i.e., for every generated graph!*

Section 3 presents a geometric graph generator that circumvents these difficulties by allowing the generator to solve the collision avoidance problem at training time, rather than once for every input.

## 3    GG-GAN: A GEOMETRIC GRAPH GENERATIVE ADVERSARIAL NETWORK

We adopt the Wasserstein GAN framework (Arjovsky et al., 2017) to learn an implicit generative model (Goodfellow et al., 2014). Our GG-GAN consists of a geometric generator $g_{gg}$ and a graph neural network discriminator $d_{gg}$, as discussed in Sections 3.1 and 3.3, respectively. GG-GAN is isomorphism consistent, scalable, and it avoids the need for collision avoidance at every input.

### 3.1    THE GEOMETRIC GRAPH (GG) GENERATOR

The generator of our GG-GAN maps a set of points from an initial configuration to a target position in a permutation equivariant manner. However, to avoid solving a collision avoidance problem for every input, we retain the initial points at *fixed*, non-random positions, while using a random *context* vector to determine the type of graph to be generated. Notice that, rather than attempting to learn to transform one distribution over sets to another, we instead learn a *distribution over functions* conditioned on a deterministic input in a manner similar to a conditional GAN — though, in contrast to the latter, we also learn the fixed input during training.

Figure 1 illustrates that to generate a graph $G$ of $n$ nodes, our generator takes as an input a fixed *learned* set $\Phi = \{\boldsymbol{\phi}_1, \ldots, \boldsymbol{\phi}_n\}$ of points in $\mathbb{R}^d$ and a vector $\boldsymbol{z}$ sampled from a Normal distribution:

$$\boldsymbol{X} = g_{gg}(\boldsymbol{\Phi}, \boldsymbol{z}), \quad \text{where} \quad \boldsymbol{z} \sim \mathcal{N}(\boldsymbol{0}, \boldsymbol{I}_q),$$

and $\boldsymbol{\Phi} \in \mathbb{R}^{n \times d}$ is the matrix representation of set $\Phi$. Then, $G$ is determined from $\boldsymbol{X}$, by sampling each edge $(v_i, v_j)$ from a Bernoulli random variable with probability $\kappa(\boldsymbol{x}_i, \boldsymbol{x}_j) = \sigma(\boldsymbol{x}_i^\top \boldsymbol{x}_j)$, where $\sigma$ is the standard sigmoid function. We stress that this approach can allow for arbitrary dependencies between edges: though any two edges become independent conditioned on a specific $\boldsymbol{x}$, the random variables can feature arbitrary dependencies overall, e.g., by setting $\kappa(\boldsymbol{x}_i, \boldsymbol{x}_j) = \kappa(\boldsymbol{x}_{i'}, \boldsymbol{x}_{j'}) = 1$ for some $\boldsymbol{z}$ and zero otherwise the two edges will always appear together.

The gradient of the non-differentiable sampling step is approximated using straight-through estimation on samples from a concrete distribution (Bengio et al., 2013; Jang et al., 2016; Maddison et al., 2016), similar to the Gumble-softmax trick used in De Cao & Kipf (2018). When applicable, one may also read out node attributes by using an MLP independently on each row of $\boldsymbol{X}$.

To generate graphs of different sizes, we model $n$ as a categorical random variable in $[n_{min}, n_{max}]$ and fix $\mathbf{\Phi}$ to contain $n$ points from $\{\phi_1, \ldots, \phi_{n_{max}}\}$. The parameters of the categorical distribution, including $n_{min}$ and $n_{max}$, are directly estimated from the training data.

To implement $g_{gg}$, we concatenate $\boldsymbol{z}$ with every $\phi_i$ and process the resulting matrix with a permutation equivariant function: $\boldsymbol{X} = g_{gg}(\mathbf{\Phi}_{\|\boldsymbol{z}})$ with $\mathbf{\Phi}_{\|\boldsymbol{z}} = [(\phi_1\|\boldsymbol{z}), \ldots, (\phi_n\|\boldsymbol{z})]^\top$. Different choices are possible on how to instantiate $g_{gg}$. For instance, $g_{gg}$ could be parametrized by a PointNetST network, which is a universal approximator over the space of equivariant functions (Segol & Lipman, 2020), or based on the constructions of Keriven & Peyré (2019) and Sannai et al. (2019). Inspired of the success of attention for set-based tasks, we use a deep self-attention network.

Let us further justify our generator using the following properties. *Expressive power:* By construction, our $g_{gg}$ can learn to model dependencies between output points, enabling it to control the probability with which each graph is generated (see Section 2.2.1). *Isomorphism consistency:* By Proposition 2, using an equivariant function $g_{gg}$ ensures that isomorphic graphs are always sampled with equal probability (see Section 2.2.2). *Collision avoidance:* Since $\Phi$ is deterministic by construction, our generator can avoid collisions by prescribing a different role to each input point $\phi_i$. There is no need to determine an assignment from $\mathbf{\Phi}$ to $\boldsymbol{X}$ every time that a graph is generated (see Section 2.2.3), the assignment can be determined once during training.

### 3.2 AN ILLUSTRATIVE TOY EXAMPLE: LEARNING TO GENERATE A TARGET SET OF POINTS

To highlight the impact of using fixed vs random points, we perform the following toy experiment.

We generate a single target $\boldsymbol{X}$ and use an equivariant neural network to determine a mapping that can transform any random set $\boldsymbol{Z}$ of points to $\boldsymbol{X}$, or specifically, $\boldsymbol{z}_i \rightarrow \boldsymbol{x}_i$. As shown in Figure 2, standard permutation equivariant architectures (an MLP applied independently on each point, PointNetST (Segol & Lipman, 2020), and multi-head attention (Bahdanau et al., 2014; Vaswani et al., 2017)) fail to learn such a mapping.

In contrast, if the random input $\boldsymbol{Z}$ becomes fixed and the randomness is conditioned to a context vector, all networks easily solve the task. An obvious caveat is that a generator using a fixed $\mathbf{\Phi}$ can learn to completely ignore the context vector and instead learn only a fixed number of graphs. We show in Appendix B that this does not happen in practice: when trained on a proper generative task, our generator indeed learns to generate different graphs for different context vectors.

Nevertheless, instead of relying on a single matrix $\mathbf{\Phi}$, it is also possible to learn a small set of such matrices and present them to the generator in batch form. In our experiments, we learn a batch of 20 such matrices. We determined this change to be beneficial in encouraging the learned model to find different $\boldsymbol{X}$ that fool the discriminator, and hence, to obtain a higher novelty.

### 3.3 THE GEOMETRIC GRAPH DISCRIMINATOR

In a Wasserstein-GAN framework, the discriminator is critical in the approximation of the Wasserstein distance. For this purpose, we use a message-passing neural network (MPNN) discriminator $d_{gg}$. Since MPNN without node attributes are known to be blind to many relevant properties of a graph's structure (Xu et al., 2019; Morris et al., 2019; Chen et al., 2020), similar to previous works (De Cao & Kipf, 2018; Yang et al., 2019), we increase the discriminative power of $d_{gg}$ by adding specially selected node and graph features. Adding node attributes can enhance expressive power, even rendering MPNN universal in the limit under a uniqueness condition (Loukas, 2019).

In particular, we build $v_i$'s attribute vector by appending to the corresponding node's degree the first $k$ entries $\boldsymbol{U}(i, : k)$ of the $i$-th row of the Laplacian eigenvector matrix, which is sorted in increasing order. Using spectral features has empirically been shown to boost performance (Dwivedi et al., 2020) and it also acts as a symmetry-breaking mechanism. We further aid the discriminator by explicitly providing it with $k$-cycle counts ($3 \leq k \leq 5$) as well as the number of nodes present in the graph currently evaluated. Cycles are a prominent graph feature that standard MPNNs cannot distinguish. Both eigenvectors and cycles are computed in a differentiable manner during the forward pass. For further details about the discriminator architecture, we refer to Appendix D.4.

Although, from a pure deep learning perspective, using handcrafted features is undesirable, to our knowledge, no universal and efficient equivariant graph classifier currently exists: MPNN is

| Models | Molgan-QM9 | | | | | CommunitySmall-20 | | | | | Chordal9 | | | | |
|---|---|---|---|---|---|---|---|---|---|---|---|---|---|---|---|
| | Deg. | Clust. | Cycle | AC | DA | Deg. | Clust. | Cycle | AC | DA | Deg. | Clust. | Cycle | AC | DA |
| MMD threshold | 0.0050 | 0.1649 | 0.1167 | 0.0784 | 0.0359 | 2.71E-5 | 0.0013 | 1.17E-8 | 0.0012 | 0.0002 | 0.0404 | 0.0090 | 0.0670 | 0.0500 | 0.1348 |
| MolGAN (RL) | 0.0054 | 0.0560 | 0.0234 | 0.1317 | 0.0342 | – | – | – | – | – | – | – | – | – | – |
| MolGAN (no RL) | 0.0045 | 0.1830 | 0.0676 | 0.1194 | 0.0866 | – | – | – | – | – | – | – | – | – | – |
| graphRNN | 0.0046 | 0.1514 | 0.1257 | 0.0675 | 0.0402 | 0.0008 | 0.1723 | 9.5E-6 | 0.0136 | 0.0676 | 0.0463 | 0.0120 | 0.0875 | 0.0492 | 0.1460 |
| Erdős-Rényi | 0.0438 | 0.3252 | 0.2221 | 0.6575 | 0.0349 | 0.0338 | 0.4421 | 0.0076 | 0.7847 | 0.4624 | 0.0204 | 0.0158 | 0.0396 | 0.0607 | 0.1304 |
| Barabasi-Albert | 0.2245 | 0.0101 | 0.3667 | 0.0980 | 0.1378 | 0.1940 | 0.4773 | 0.0003 | 0.9799 | 1.3517 | 0.0375 | 0.0138 | 0.0274 | 0.2627 | 0.0074 |
| ScrMatch | 0.0638 | 0.1689 | 0.0606 | 0.1412 | 0.0944 | 0.0034 | 0.4181 | 0.0004 | 0.5940 | 0.3152 | 0.2032 | 0.0603 | 0.2929 | 0.1518 | 0.9974 |
| CondGEN | 0.0954 | 0.3970 | 0.5427 | 1.2002 | 0.1485 | 0.1497 | 0.4759 | 0.0180 | 1.1582 | 1.2712 | 0.0325 | 0.0095 | 0.0147 | 0.5522 | 0.0342 |
| PointMLP-GAN | 0.0126 | 0.0614 | 0.0350 | 0.1059 | 0.0602 | 0.0254 | 0.4594 | 0.0015 | 0.8104 | 0.3862 | 0.0260 | 0.0524 | 0.0795 | 0.1235 | 0.2988 |
| GG-GAN (RS) | 0.0143 | 0.0145 | 0.0255 | 0.2992 | 0.0084 | 0.0354 | 0.4339 | 0.0001 | 0.6755 | 0.3968 | 0.0267 | 0.0110 | 0.0219 | 0.1864 | 0.2487 |
| **GG-GAN** | 0.0043 | 0.0703 | 0.0152 | 0.0338 | 0.0065 | 0.0080 | 0.4258 | 0.0003 | 0.2553 | 0.3454 | 0.0391 | 0.0145 | 0.0329 | 0.0411 | 0.1659 |

Table 1: Comparing GG-GAN to SotA methods in graph generation. The reported scores (*lower is better*) measure the MMD between the graph statistics of 512 test and generated graphs. The MMD threshold measures differences between training and test sets; it aims to quantify how small a "good" score should be. Green indicates a score not larger than the MMD threshold, and larger scores are annotated with a color gradient from yellow to red (in regular increments up to a max MMD of 1).

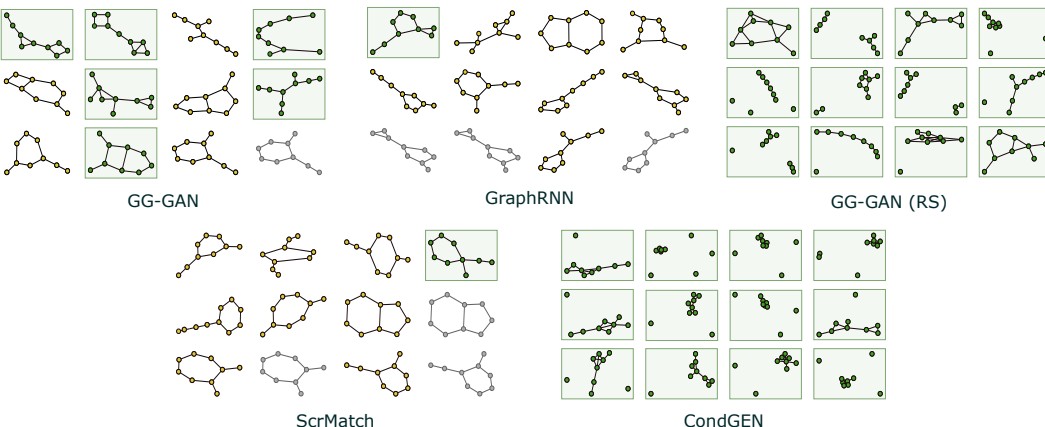

Figure 3: Examples (non-handpicked) of generated graphs on the QM9 dataset. Novel graphs are in green, graphs found in the training set are in orange, and duplicates are grayed out. GG-GAN exhibits higher novelty than all methods that capture the QM9 distribution (see also Figures 6 and 7).

not universal, adding node attributed breaks equivariance, and because more powerful equivariant classifiers manipulate higher-order tensors (Maron et al., 2019; Morris et al., 2019; Vignac et al., 2020) they carry increased computational and memory requirements. We expect the further advances on GNNs to improve our framework and to eliminate the need for handcrafted features.

# 4 NUMERICAL EVIDENCE

We evaluate GG-GAN's performance to correctly model the distribution statistics as well as to generate novel high-quality graphs. We proceed in three veins, testing how well generators can *model complex dependencies*, how well generators can *scale*, and how well generators can identify *novel* objects, respectively. Consistent with previous works, we focus on the QM9 ($n = 9$) and Community ($n = 20$) datasets. We also introduce the Chordal ($n = 9$) dataset. All three datasets are of small size, which allows us to compare even with the slowest baselines, whose descriptions are in Appendix C.2.

## 4.1 MODELLING COMPLEX DEPENDENCIES

Building on the evaluation protocol of You et al. (2018), we computed the maximum mean discrepancy (MMD) over 5 standard graph statistics defined in Appendix C.3: *degree distribution* (Deg.), *cycle distribution* (Cycle), *clustering* (Clust.), *algebraic connectivity* (AC), and *degree assortativity*

| Models | Molgan-QM9 | CommunitySmall-20 | Chordal9 |
|---|---|---|---|
| | isomorphism classes | isomorphism classes | isomorphism classes |
| MolGAN | 318 | – | – |
| MolGAN (no RL) | 182 | – | – |
| graphRNN | 132 | 4200 | 178 |
| ScrMatch | 101 | **5000** | 12 |
| PointMLP-GAN | 26 | 1474 | 13 |
| GG-GAN | **832** | **5000** | **232** |

Table 2: Number of isomorphism classes not in the training set found within a sample of 5000 generated graphs (*higher is better*). We report novelty for generators producing graphs with reasonable MMD scores. In Chordal9, we verify whether the generated graph is chordal and different from the training set (rather than a direct comparison with the test set).

*coefficient* (DA). We argue that a good generative model should generally score sufficiently well across *all* MMDs. In our experience, even failing to capture a single statistic can lead to generating graphs with significant visual differences from the dataset. To specify what "sufficient well" means, we set as threshold the MMD between the training and test sets: If a generator is close to this threshold, then we say that it captures well a given graph statistic.

Table 1 summarizes our results. As observed, GG-GAN is the model with the most MMD values below the defined threshold, scoring below or equal to the threshold in 8 out of 15 cases (with the second best model achieving good MMDs in 5 out of the 15 cases). This indicates GG-GAN can model well both simple and complex graph statistics. Intriguingly, no model can achieve results below the threshold in the Community dataset, although such dataset is simple and can, theoretically, be modeled exactly with an MLP acting independently on each point (PointMLP-GAN).

In fact, in the Community dataset, the latter achieves poorer performance than GG-GAN, and both are surpassed by graphRNN, perhaps since the latter is trained using a likelihood-based loss (rather than implicitly as GANs do) which fits well to a random graph with communities. GraphRNN's good overall MMD scores agree with our intuition that autoregressive approaches are capable of modeling complex structural dependencies. On the other hand, MolGAN is generally apt at generating molecules, but its performance can change when the custom reinforcement learning (RL) loss is not utilized. Moreover, MolGAN's specialized architecture is not applicable to non-molecular data.

To illustrate the effect of collision avoidance, we also compare GG-GAN with an ablated random set (RS) version. The latter is identical to GG-GAN in all aspects except that the input set contains points sampled from a Normal distribution (i.e., rather than having a fixed set and a random context). As seen, though the RS version can model excellently simple statistics (like DA and Clust.), it struggles with more complex ones, such as AC. Figure 3 visually demonstrates the importance of reproducing all graph statistics (rather than some): whereas the graphs generated by GG-GAN RS differ from those in the dataset, GG-GAN manages to capture both local and global topological properties.

## 4.2 GENERATION NOVELTY

We verify how well the trained generative models can synthesize unique graphs. Specifically, we study the number of isomorphism classes generated that differ from those in training set. Table 2 shows that most models struggle to attain a high novelty score within a sample of 5k graphs. We see that the majority of generated graphs either belong to a small set of isomorphism classes or duplicate the training dataset. GraphRNN, in particular, does not exhibit high novelty in our experiments. This provides evidence that the low MMD scores of graphRNN can (at least partially) be attributed to learning-to-memorize the training set. Perhaps due to not being isomorphism consistent, MolGAN achieves low novelty, though we observe a boost via the RL module.

It is also important to note that attaining a high novelty score is challenging only for methods that can closely fit MMD statistics—clearly, it can be trivial to generate novel graphs, if they share few things in common with the distribution of interest. For this reason, we left CondGEN and GG-GAN (RS) out from the comparison—as seen in Figure 3, most graphs that these methods generated in our experiments statistically and perceptually stood out significantly from the training distribution. On the other hand, GG-GAN's novelty surpasses or matches that of all other methods, while simultaneously achieving competitive MMD scores.

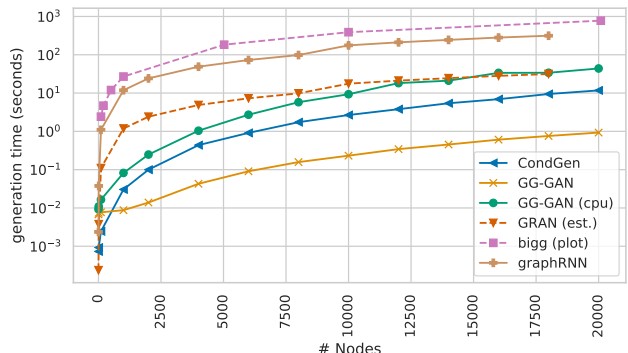

Figure 4: Median generation times for individual graphs of increasing size (lower is better). Dashed lines indicate that the timings are based on reports of previous papers. GG-GAN outperforms autoregressive baselines and the non-autoregressive CondGen by one or more orders of magnitude.

### 4.3 SCALABILITY

Since real life graphs often have orders of thousands or even millions of nodes, the scalability of any generative method is critical. Specifically, graphRNN, GRAN, and bigg (You et al., 2018; Liao et al., 2019; Dai et al., 2020) aim to push the previous SotA in terms of graph size.

We test our method against CondGen and graphRNN on the same machine we ran our models on and (due to time constraints) estimate the GRAN timing based on the speedup reported[2] in Liao et al. (2019). For bigg, we extracted their reported timings from their figure 6 using WebPlotDigitizer (Rohatgi, 2020). We perform all measurements by initialising the model and then generating a number of $N$ random graphs ($N$=100 for ours, $N$=20 for graphRNN) using an untrained model. We report sizes until $n = 20k$, at which point our model runs out of memory.

Figure 4 demonstrates that our model is highly efficient, outperforming graphRNN, the estimated GRAN, and even bigg in terms of inference speed even when it is run on CPU. CondGen is competitive with our model on CPU (the publicly available version actually performs the conversion of embeddings into a graph on CPU) which is not surprising since it is also a non-autoregressive model.

We also highlight that the released version of CondGen has over $100\times$ less parameters than our model. Nevertheless, since ConGen also non-autoregressive and uses an outer-product kernel to generate adjacency matrices, in principle it scales with $\mathcal{O}(n^2)$, similar to GG-GAN. It might be interesting to note that Dai et al. (2020) in theory scales better than our method due to impressive optimizations exploiting sparsity, leading to $\mathcal{O}((m + n) \log n)$ scaling for $n$ nodes with $m$ edges. However, for the graph sizes we tested, the constant factor incurred by either their framework or their implementation seems to be significant enough that we still outperform them in terms of generation speed. We also highlight that the superior speed of GG-GAN is a consequence of our generator's parallelizability and not a consequence of specialized optimization and engineering.

## 5 CONCLUSIONS

Our work partakes in the geometric graph generation research thread and advocates for learning a distribution over functions rather than transforming a distribution over points. Our methodology yields an isomorphism consistent generator that, as empirically confirmed, possesses favorable scalability, novelty, and expressive power, matching or surpassing various SotA baselines in most cases.

A key direction that we would like to focus in the future is scalability. Herein, we rely on the multi-head attention of Vaswani et al. (2017) that scales with $O(n^2)$ in compute and memory. More recent work (Wang et al., 2020; Shen et al., 2018; Katharopoulos et al., 2020) has reduced this complexity to nearly $O(n)$ while maintaining performance, implying low hanging fruit in additional efficiency gains. We are also interested in more sophisticated generator architectures that exploit sparsity, possibly using insights from Dai et al. (2020).

---

[2]They report a $6\times$ speedup w.r.t to graphRNN at quality parity on a GTX 1080Ti. We round this up to a $10\times$ speedup in an attempt to be as fair as possible (to account for our more modern hardware).

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

# A  DEFERRED PROOFS

## A.1  PROOF OF THEOREM 1

For completeness, before presenting the proof, we restate the Theorem of Maehara and Rödl showing that $k = 2\Delta$ dimensions suffice (Maehara & Rödl, 1990):

**Theorem 2** (Maehara and Rödl (Maehara & Rödl, 1990))**.** For every undirected *simple* graph $G$ of degree at most $\Delta$, there exist points $\boldsymbol{x}_1, \ldots, \boldsymbol{x}_n$ in $\mathbb{R}^k$ with $k = 2\Delta$, for which $\kappa(\boldsymbol{x}_i, \boldsymbol{x}_j) = 1$ if and only if $(v_i, v_j) \in E$, i.e., $\kappa(\boldsymbol{x}_i, \boldsymbol{x}_j) = \mathbf{1}_{\{\boldsymbol{x}_i^\top \boldsymbol{x}_j = 0\}}$, where $\mathbf{1}_C$ is the indicator function of set $C$.

We move on to our proof. Let $q$ be a function that is strictly increasing and $\lambda$-Lipschitz. We consider the matrix $\boldsymbol{C}$ with entries given by the following expression:

$$\boldsymbol{C}(i,j) = q^{-1}(\boldsymbol{A}(i,j) + \eta) \quad \text{and} \quad \boldsymbol{A}(i,j) = q(\boldsymbol{C}(i,j)) - \eta$$

for $\eta = q(0)$. If $\boldsymbol{A}(i,j) \in [0,1]$, we then also have

$$0 = q^{-1}(\eta) = \min_{a \in [\eta, 1+\eta]} q^{-1}(a) \le \boldsymbol{C}(i,j) \le \max_{a \in [\eta, 1+\eta]} q^{-1}(a) = q^{-1}(1+\eta) = q^{-1}(1+q(0)) = c.$$

Denote by $\boldsymbol{U}\boldsymbol{\Lambda}\boldsymbol{U}^\top$ the eigenvalue decomposition of $\boldsymbol{C}$ and write

$$\boldsymbol{C} = \boldsymbol{X}\boldsymbol{X}^\top - \gamma\boldsymbol{I}, \quad \text{with} \quad \boldsymbol{X} = \boldsymbol{U}(\boldsymbol{\Lambda} + \gamma\boldsymbol{I})^{1/2} \in \mathbb{R}^{n \times n}$$

and $\gamma = \|\boldsymbol{C}\|_2$. This is always possible since $\boldsymbol{U}(\boldsymbol{\Lambda} + \gamma\boldsymbol{I})^{1/2}(\boldsymbol{\Lambda} + \gamma\boldsymbol{I})^{1/2}\boldsymbol{U}^\top = \boldsymbol{U}(\boldsymbol{\Lambda} + \gamma\boldsymbol{I})\boldsymbol{U}^\top = \boldsymbol{U}\boldsymbol{\Lambda}\boldsymbol{U}^\top + \gamma\boldsymbol{I} = \boldsymbol{C} + \gamma\boldsymbol{I}$ is, by the choice of $\gamma$, positive semi-definite.

If we think of the rows $\boldsymbol{x}_1, \ldots, \boldsymbol{x}_n$ of $\boldsymbol{X}$ as points, we have

$$\|\boldsymbol{x}_i\|_2^2 \le \max_i \boldsymbol{\Lambda}(i,i) + \gamma \le 2\|\boldsymbol{C}\|_2 \le 2\max_i \sum_{j=1}^n |\boldsymbol{C}(i,j)| = 2\max_i \sum_{j=1}^n |q^{-1}(\boldsymbol{A}(i,j) + \eta)| \le 2c\Delta.$$

To proceed, we rely on the following well-known property (Vempala, 2005): If $\bar{\boldsymbol{x}}_i = \boldsymbol{R}\boldsymbol{x}_i$ and $\boldsymbol{R}$ is a $k \times n$ random matrix whose entries are independently sampled from $\mathcal{N}(0, \frac{1}{\sqrt{k}})$, then for any $n$ points $\boldsymbol{x}_1, \ldots, \boldsymbol{x}_n \in \mathbb{R}^n$ with $\|\boldsymbol{x}_i\|_2^2 \le \delta$, we have

$$P(|\boldsymbol{x}_i^\top \boldsymbol{x}_j - \bar{\boldsymbol{x}}_i^\top \bar{\boldsymbol{x}}_j| \ge t\,\delta) < 4n^2 \exp\left(-\frac{(t^2 - t^3)\,k}{4}\right) \quad \text{for all} \quad i, j = 1, \ldots, n.$$

It follows by the probabilistic argument, that choosing $k = \frac{8\log(2n)}{\phi(t)}$ with $\phi(t) = t^2 - t^3$ suffices for some points $\boldsymbol{x}_1, \ldots, \boldsymbol{x}_n$ to exist in $\mathbb{R}^k$ for which $|\boldsymbol{x}_i^\top \boldsymbol{x}_j - \bar{\boldsymbol{x}}_i^\top \bar{\boldsymbol{x}}_j| \le t\delta$.

In this manner, we work backwards and consider the matrix with elements $\bar{\boldsymbol{C}}(i,j) = \bar{\boldsymbol{x}}_i^\top \bar{\boldsymbol{x}}_j - \gamma$, which approximates $\boldsymbol{C}$ as follows:

$$\|\boldsymbol{C} - \bar{\boldsymbol{C}}\|_\infty = \|\boldsymbol{X}\boldsymbol{X}^\top - \bar{\boldsymbol{X}}\bar{\boldsymbol{X}}^\top\|_\infty \le (2c\Delta)\,t.$$

For the matrix with elements $\bar{\boldsymbol{A}}(i,j) = q(\bar{\boldsymbol{C}}(i,j)) - \eta$, we have

$$\|\boldsymbol{A} - \bar{\boldsymbol{A}}\|_\infty = \max_{i,j} \|q(\boldsymbol{C}(i,j)) - \eta - \left(q(\bar{\boldsymbol{C}}(i,j)) - \eta\right)\|_\infty \le \lambda\|\boldsymbol{C} - \bar{\boldsymbol{C}}\|_\infty \le t\,(2\lambda c\Delta).$$

Selecting $t = \frac{\epsilon}{2\lambda c\Delta}$ then yields $k = \frac{8\log(2n)}{\phi(\epsilon/(2\lambda c\Delta))}$ and $\|\boldsymbol{A} - \bar{\boldsymbol{A}}\|_\infty \le \epsilon$.

Let us now consider the shifted exponential function $\kappa(\boldsymbol{x}_i, \boldsymbol{x}_j) = e^{\boldsymbol{x}_i^\top \boldsymbol{x}_j} - 1$. We set $q(x) = e^x$, for which $\eta = q(0) = 1$ and $c = q^{-1}(1 + \eta) = \log(2)$. Moreover, within the domain $[0, c]$ of interest, the Lipschitz constant of $q$ is given by $\lambda = e^c = 2$. Substituting the above to the bound gives $k = \frac{8\log(2n)}{\phi(\epsilon/(4\log(2)\Delta))}$. The proof concludes by noting that function $\phi$ is lower bounded as $t^2$ in the domain of interest.

## A.2  Proof of Corollary 1

The proof follows trivially by selecting $\mathcal{D}_x$ to have uniform measure on the $n$ points that give rise to the graph according to Theorem 1. A more detailed analysis of the probability in question can be found in the proof of Proposition 1.

## A.3  Proof of Proposition 1

Let $X = \{x_1, \ldots, x_n\}$ be the set that we wish to generate and denote by $\mathcal{D}$ the distribution in $\mathbb{R}^k$ that the random generator is induced from (with $m = k$). Write also as $p_i = \text{Prob}(x_i \sim \mathcal{D})$ the probability that the $i$-th point is drawn from $\mathcal{D}_x$.

The probability that $X$ is sampled is given by

$$\text{Prob}(X \sim \mathcal{D}_x) = \sum_{\pi \in \mathfrak{S}_n} p_{\pi_1} \cdots p_{\pi_n} = n! \prod_{i=1}^n p_i,$$

where $\pi$ is an element of the permutation group $\mathfrak{S}_n$ on $n$ elements. The above (log) product is bounded by

$$\log\left(\prod_{i=1}^n p_i\right) = n \sum_{i=1}^n \frac{1}{n} \log p_i \le n \log\left(\sum_{i=1}^n \frac{1}{n} p_i\right) \le n \log\left(\frac{1}{n}\right),$$

where the first inequality follows by Jensen and the second is because $\sum_i p_i \le 1$. Thus, employing Stirling's approximation we obtain

$$\begin{aligned}
\log\left(\text{Prob}(X \sim \mathcal{D}_x)\right) &\le \log(n!) - n \log n \\
&\le n(\log n - 1) + O(\log n) - n \log n \\
&= -n + O(\log n),
\end{aligned}$$

which implies that there exists a constant $c$ for which $\text{Prob}(X \sim \mathcal{D}_x) \le cne^{-n}$, as needed.

## A.4  Proof of Proposition 2

To prove the proposition it suffices to show that $\text{Prob}(\pi X) = \text{Prob}(X)$ for any permutation $\pi$ and any $X = g(Z)$.

The probability of any $X$ to be sampled can be expressed as

$$\text{Prob}(X) = \int 1(g(Z) = X) \text{Prob}(Z) \, dZ,$$

where $1(g(Z) = X)$ is the indicator function of event $g(Z) = X$ and the integration is over all $Z$. Similarly, we have that

$$\begin{aligned}
\text{Prob}(\pi X) &= \int 1(g(Z') = \pi X) \text{Prob}(Z') \, dZ' \\
&= \int 1(g(\pi Z) = \pi X) \text{Prob}(\pi Z) \, dZ & \text{(by the substitution } Z' = \pi Z) \\
&= \int 1(g(\pi Z) = \pi X) \text{Prob}(Z) \, dZ & \text{(since } \text{Prob}(Z) = \text{Prob}(\pi Z)) \\
&= \int 1(g(Z) = X) \text{Prob}(Z) \, dZ & \text{(} g \text{ is permutation equivariant)} \\
&= \text{Prob}(X),
\end{aligned}$$

where the second equality is true since $Z' = \pi Z$ is an isometric mapping and $1(g(\cdot) = \pi X) \text{Prob}(\cdot)$ is a real-valued function.

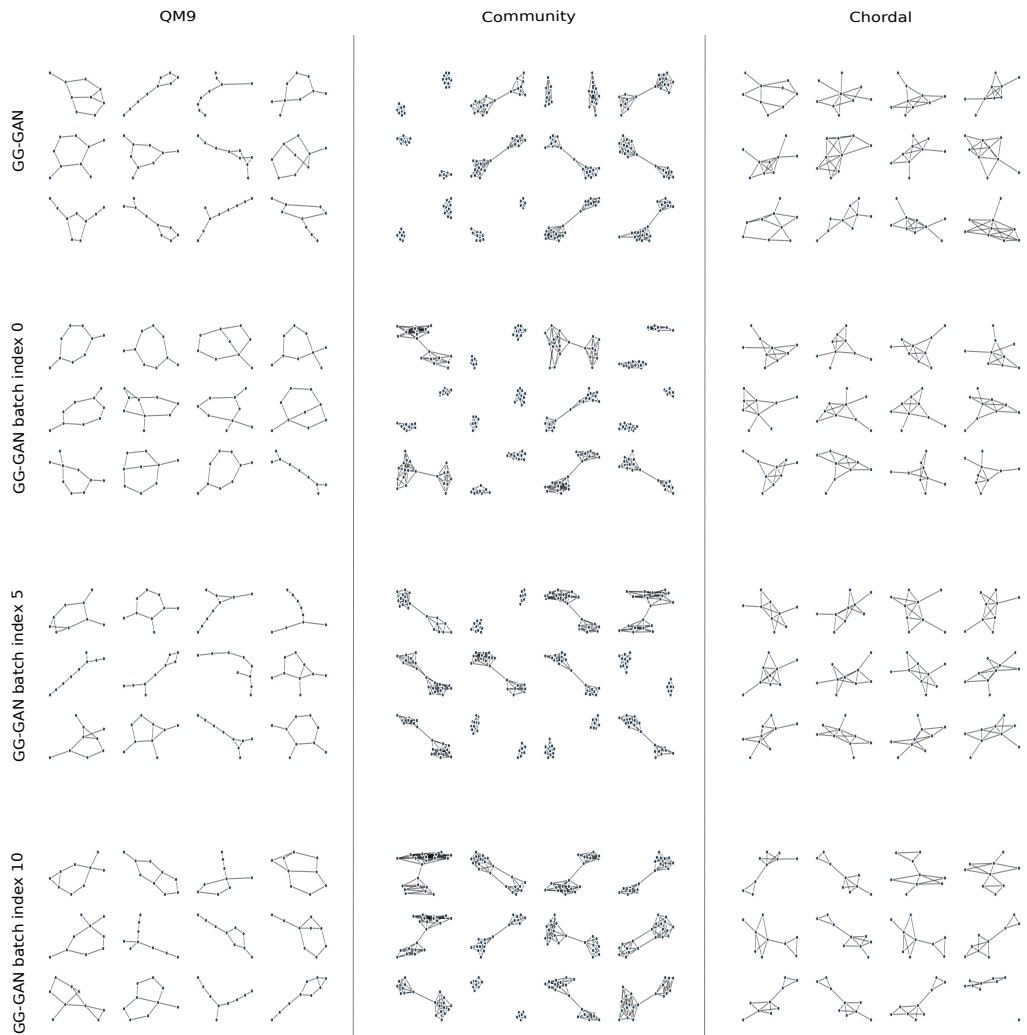

Figure 5: Random sampled graphs from GG-GAN, and GG-GAN at specific batch indices.

## B    BATCH INDEX DIVERSITY

As we discussed in the main text, GG-GAN learns a function distribution operating on a fixed batch of point sets. One possible failure mode of this approach is that the generator might learn to ignore the randomly drawn context and simply learn a single fixed mapping for each batch index of the fixed embedding.

To confirm that this does not happen in practice, we sampled graphs from our generator and checked for diversity within the samples drawn from each batch index (0, 5, 10) across the samples. The results are shown in Figure 5. Notice that if the generator were to ignore the context vector, we would expect no diversity on each batch index (i.e., for the same fixed $\Phi$) across samples. However, as can be seen in the figure, for the QM9, at batch index 0 a circle structure appears to be learned, but overall there is diversity within samples in the same batch for all datasets.

## C EXPERIMENTAL DETAILS

### C.1 HYPERPARAMETERS

The hyperparameters used in our experiments can be found in Table 3. The baselines were run with code provided in the respective papers, see Appendix D.1. ExtraAdam Gidel et al. (2018) was used for both generator and discriminator with the following parameters: `lr=1e-4, betas=(0.5, 0.9999), eps=1e-8, weight_decay=1e-3, ema=False, ema_start=100,` using the standard 5:1 ratio of discriminator to generator updates.

| Params. | Molgan-QM9 | | CommunitySmall-20 | | Chordal9 | |
|---|---|---|---|---|---|---|
| | PointMLP | GG-GAN | PointMLP | GG-GAN | PointMLP | GG-GAN |
| Batch size | 20 | 20 | 20 | 20 | 20 | 20 |
| Attention layers | – | 6 | – | 3 | – | 3 |
| $\Phi$ size | 50 | 25 | 50 | 25 | 50 | 25 |
| Context vector size | 50 | 5 | 50 | 25 | 50 | 25 |
| MLP layers | [128,256,512] | – | [128,256,512] | – | [128,256,512] | – |
| Optimizer | ExtraAdam | ExtraAdam | ExtraAdam | ExtraAdam | ExtraAdam | ExtraAdam |

Table 3: Hyperparameters used in the experiments for PointMLP and GG-GAN

Table 4 summarizes the hyperparameters used for our discriminator (see Table D.4 for an explanation of the architecture) throughout all our experiments. $L$ refers to the number of ResBlocks in the DenseGIN, $c_h^r$ is the hidden layer of the final readout MLP, $c_o^l$ is the output node feature dimension of the ResBlock $l$, $c_i^l$ and $c_h^l$ analogously refer to the input features of each ResBlock and the width of the hidden layer of the SkipGINs inner MLP respectively.

| $L$ | $c_h^r$ | ResBlock Parameters |
|---|---|---|
| 7 | 128 | $c_i^1 = 9, c_h^l = 64, c_o^l = [32, 64, 64, 64, 128, 128], c_i^l = c_o^{l-1}$ |

Table 4: Discriminator hyperparameters for all experiments

### C.2 DATASETS

**QM9**. The QM9 dataset contains 133,885 organic compounds. Following De Cao & Kipf (2018), our experiments focused on a 5k subset. Most of the graphs in this dataset have 9 nodes.

**Chordal**. A graph is chordal if every cycle of length at least 4 has an edge connecting two nodes of the cycle, but is not of the cycle itself. This dataset (McKay, 2020) contains 11911 chordal graphs of size 9, from which we also use a 5000 subset for training and the rest 6911 for test.

**Community**. Similarly to Niu et al. (2020), we construct 5000 $N=20$ node graphs containing 2 communities generated by the Erdŝ-Rényi model Erdös (1959) with $p = 0.7$. Then, $0.05N$ edges are added between the communities.

### C.3 MODELING COMPLEX DEPENDENCIES

We first define the five discrete distributions that the MMD scores are based on:

- *Degree distribution*. The degree of a node is the number of connections it has to other nodes. The degree distribution is the probability distribution of these degrees over the entire graph.
- *Cycle distribution*. A cycle is a non-empty sequence of adjacent nodes in which the only repeated vertices are the first and last vertices. The cycle (length) distribution is the probability distribution of the lengths of all cycles within a graph.
- *Clustering*. The clustering coefficient of a node is the fraction of all possible triangles through that node that exist in the current graph. The clustering (coefficient) distribution is the probability distribution of the clustering coefficients of all nodes within a graph.

- *Algebraic connectivity*. Algebraic connectivity refers to the second smallest eigenvalue of the combinatorial Laplacian matrix of the graph.

- *Degree assortativity coefficient*. The assortativity coefficient is the Pearson correlation coefficient of degree between pairs of linked nodes.

| Test sets | Molgan-QM9 | | | | | CommunitySmall-20 | | | | | Chordal9 | | | | |
|---|---|---|---|---|---|---|---|---|---|---|---|---|---|---|---|
| | Deg. | Clust. | Cycle | AC | DA | Deg. | Clust. | Cycle | AC | DA | Deg. | Clust. | Cycle | AC | DA |
| Test set 1 | 0.0049 | 0.07404 | 0.1212 | 0.0772 | 0.04236 | 2.36E-05 | 2.64E-09 | 9.01E-10 | 0.0002 | 0.0002 | 0.0404 | 0.0154 | 0.0670 | 0.0500 | 0.1348 |
| Test set 2 | 0.0056 | 0.1073 | 0.1171 | 0.0880 | 0.0306 | 8.94E-06 | 2.63E-08 | 9.41E-11 | 0.0033 | 0.0002 | – | – | – | – | – |
| Test set 3 | 0.0052 | 0.0838 | 0.1118 | 0.0699 | 0.0347 | 4.88E-05 | 3.60E-09 | 3.42E-08 | 0.0001 | 0.0003 | – | – | – | – | – |

Table 5: MMD results between the training set and the test sets used.

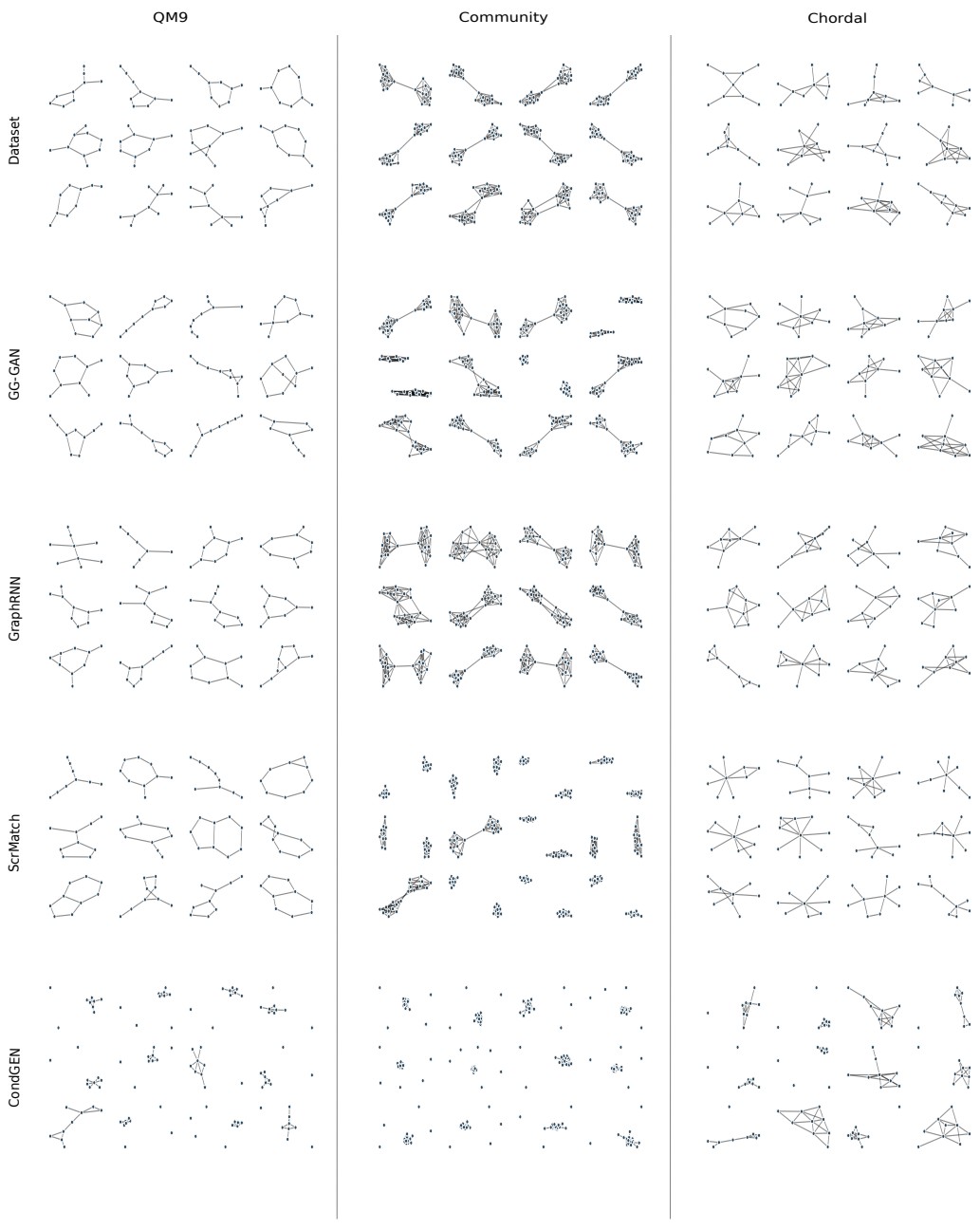

Figure 6: Random sampled graphs from the models and training dataset.

| Models | Deg. | Clust. | Cycle | AC | DA | Graphs in Dataset | Iso. classes |
|---|---|---|---|---|---|---|---|
| | | | | Molgan-QM9 | | | |
| GG-GAN | 0.0063±0.0029 | 0.0763±0.0045 | 0.0158±0.0009 | 0.0226±0.0124 | 0.0212±0.0104 | 1465.67±134.40 | 721.0±232.41 |
| GG-GAN RS | 0.0196±0.0029 | 0.0181±0.0042 | 0.0307±0.0035 | 0.3689±0.0462 | 0.0122±0.002 | 219.0±17.68 | 2046.0±20.99 |
| PointMLP | 0.0104±0.0032 | 0.0261±0.0051 | 0.0403±0.0098 | 0.0598±0.0328 | 0.057±0.027 | 1667.0±423.42 | 20.33±1.70 |
| PointMLP RS | 0.004±0.004 | 0.0405±0.0114 | 0.0525±0.0026 | 0.1954±0.005 | 0.0239±0.0033 | 474.33±22.69 | 1451.67±17.44 |
| | | | | CommunitySmall-20 | | | |
| GG-GAN | 0.0403±0.0255 | 0.4439±0.0128 | 0.0023±0.0019 | 0.491±0.2185 | 0.5085±0.173 | 0.00±0.00 | 4729.33±382.78 |
| GG-GAN RS | 0.0593±0.0228 | 0.4386±0.012 | 0.0005±0.0004 | 0.5757±0.1917 | 0.5487±0.149 | 0.00±0.00 | 4999.67±0.47 |
| PointMLP | 0.0821±0.0961 | 0.4395±0.0096 | 0.0053±0.006 | 0.7704±0.0688 | 0.3754±0.1071 | 0.00±0.00 | 2286.33±710.39 |
| PointMLP RS | 0.0721±0.0494 | 0.4405±0.0074 | 0.0136±0.005 | 0.8058±0.011 | 0.2969±0.0936 | 0.00±0.00 | 5000±0 |
| | | | | Chordal9 | | | |
| GG-GAN | 0.0385±0.0007 | 0.0187±0.0054 | 0.0504±0.0129 | 0.0479±0.005 | 0.1895±0.0227 | 2471.33±347.98 | 248.67±135.49 |
| GG-GAN RS | 0.0284±0.0044 | 0.013±0.0018 | 0.0237±0.0049 | 0.2301±0.0332 | 0.2232±0.025 | 1003.33±79.81 | 724.0±74.25 |
| PointMLP | 0.0938±0.0027 | 0.0702±0.0105 | 0.0861±0.0418 | 0.1978±0.0407 | 0.0826±0.0201 | 462.67±212.57 | 6.67±5.44 |
| PointMLP RS | 0.0453±0.0035 | 0.0165±0.0006 | 0.028±0.0018 | 0.1705±0.0143 | 0.2347±0.0108 | 313.0±15.25 | 856.33±26.78 |

Table 6: Comparison between random context (RC) and random set (RS) models. The mean and standard deviation correspond to the results of 3 runs of the same model. We also report the generated graphs in the dataset and the isomorphic classes of such models.

| | MolGAN | MolGAN (no RL) | graphRNN | ScrMatch | CondGEN | PointMLP-GAN | GG-GAN | GG-GAN RS |
|---|---|---|---|---|---|---|---|---|
| Molgan-QM9 | 318 | 182 | 132 | 101 | 660 | 26 | 832 | 2046 |
| CommunitySmall-20 | – | – | 4200 | 5000 | 5000 | 1474 | 5000 | 5000 |
| Chordal9 | – | – | 178 | 12 | 1632 | 13 | 232 | 724 |

Table 7: Number of isomorphism classes not in the training set found within a sample of 5000 generated graphs (*higher is better*). In Chordal9, we additionally verify whether the generated graph is chordal or not (rather than a direct comparison with the test set).

## C.4 SCALABILITY EXPERIMENT DETAILS

Each method was used to generate a single graph for $N$-times ($N = 100$ for Condgen and GG-GAN, 20 for graphRNN) and the median was reported. CPU evaluation done with an Intel Xeon Gold 6240, GPU evaluation done on a Nvidia V100 on the same machine. GraphRNN was run with a context of 20 previously generated nodes. We observed roughly linear scaling decrease in nodes per second as we increased the context $n_c$, e.g. with $n_c = 9$ we could generate $\approx 90$ nodes per second, with $n_c = 20$ this dropped to $\approx 47$ .

## D IMPLEMENTATION DETAILS

### D.1 EXTERNAL BASELINES

All our models are implemented and trained using pytorch and pytorch-lightning (Paszke et al., 2019; Falcon, 2019), with experiments tracked using sacred (Klaus Greff et al., 2017). For the external baselines we used the following implementations, with the author-provided hyperparameters:

1. MolGAN (De Cao & Kipf, 2018)

2. CondGen (Yang et al., 2019)

3. graphRNN (You et al., 2018)

4. ScoreMatching (Niu et al., 2020)

### D.2 WGAN

We follow the WGAN-GP approach (Gulrajani et al., 2017) except using the LP penalty from Petzka et al. (2018). In order to penalize the discriminator we calculate the penalty on convex combination between the real and fake samples (both nodes and adjacency matrices) using a uniformly sampled interpolation coefficient as is common practice.

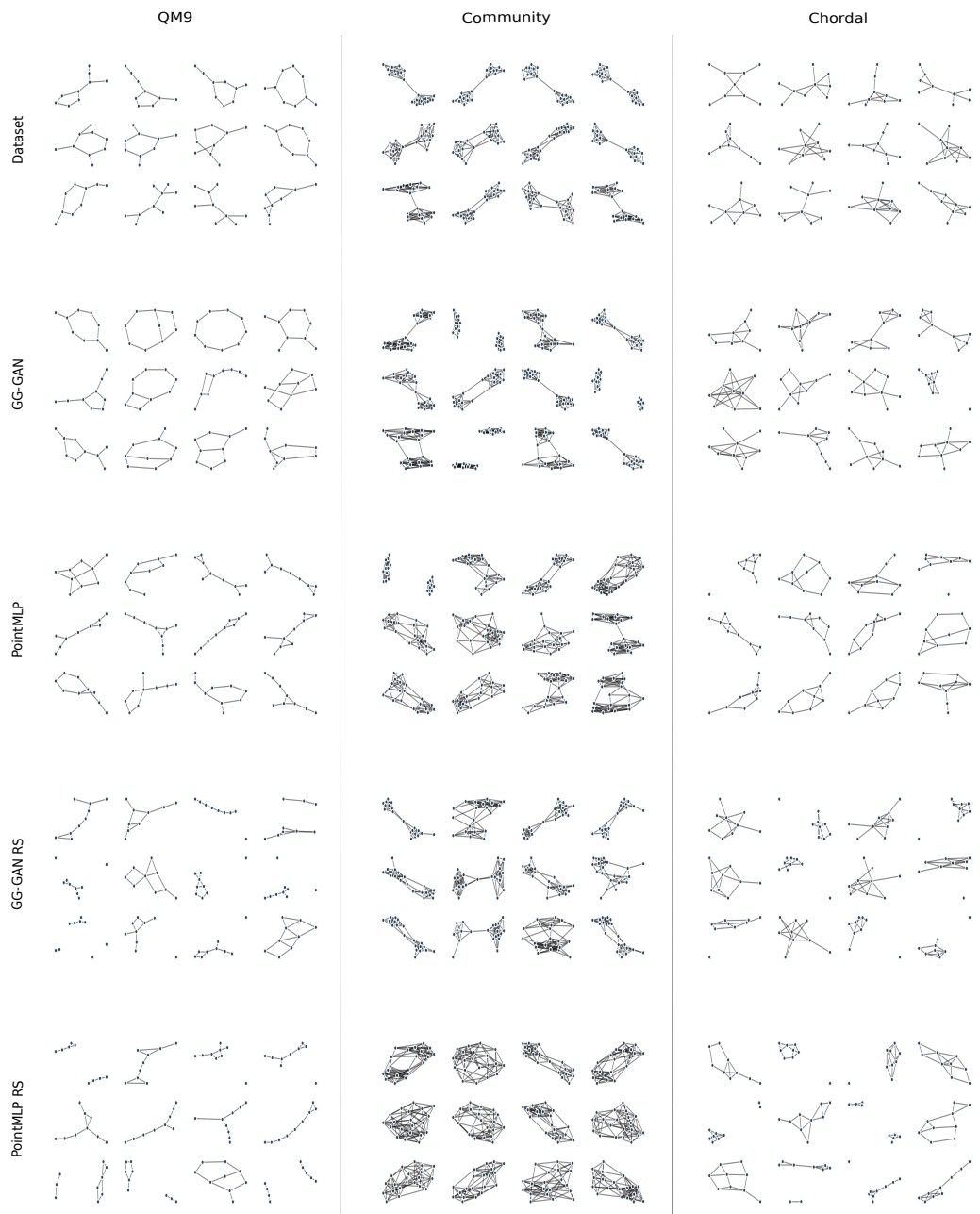

Figure 7: Comparison between random context (RC) and random set (RS) models.

## D.3 GENERATOR

In the GG-GAN model, the generator is composed of multi-head attention layers, usually between 3 and 6 (see table 3), also with skip connections after each layer and instance normalization. For the discretization of the adjacency matrix, we sample each edge from a Bernoulli random variable, as explained in Section 3.1. Afterwards, we zero the diagonal of the matrix and symmetrize it.

For the PointMLP-GAN model, our generator which outputs the generated points $x_i$ is composed of a 3 layer MLP ([128,256,412]) with ReLU activations, skip connections at each layer, and instance normalization between them. The node readout function is a simple linear layer. Similar to GG-GAN, the edge readout function is $\sigma(x_i^\top x_j)$, where $\sigma$ is the standard sigmoid function.

### D.4  DISCRIMINATOR

Since for a WGAN we require a powerful discriminator, we started combined insights from multiple papers. We start with the GIN (Xu et al., 2019) message passing block as a strong baseline using the implementation from Fey & Lenssen (2019), then build a model inspired by DenseNet (Huang et al., 2018) to increase the networks parameter efficiency and gradient y2low. We first create a primitive which we call SkipGIN: in addition to the standard GIN update prodecure, we add the output of a Linear Transmission (LT) layer as in Segol & Lipman (2020) without non-linearity which serves as a permutation equivariant skip connection. We then form residual blocks (ResBlock) out of the SkipGIN layers, where each block acts as a pre-activation residual block (He et al., 2016) using instance normalization (Ulyanov et al., 2017)[3], ReLU activation[4] and a SkipGIN in the residual path and a completely linear SkipGIN as the projection. Finally, the output of each of these blocks is collected and concatenated along the feature dimension before being passed into the readout layer. This architecture is somewhere between a full pre-activation ResNet and a Densenet, allowing the readout to access features from different depths while also ensuring gradient flow through all components. We refer to this GNN architecture as DenseGIN.

The readout layer is a single hidden layer MLP which in addition to the sum of concatenated node features from the DenseGIN receives: the 3,4,5 and 6 cycle counts and maps this to the final score, again using a ReLU nonlinearity.

---

[3]With statistics computed on each node individually in order to avoid correlation in the gradient penalty as discussed in Gulrajani et al. (2017)

[4]We also experimented with Swish but found no improvement in performance.

