# OpenReview forum: "GG-GAN: A Geometric Graph Generative Adversarial Network"
_ICLR.cc/2021/Conference — Reject_

### Official Review · AnonReviewer1 · 2020-10-28
**A good paper, some parts may need to be clarified.**

**Rating:** 7
**Confidence:** 4

**Review:**

The paper investigates graph generation using adversarial technics. They introduce an algorithm named GG-GAN, based on Wassertain GAN, in order to accurately generates new graphs in hopefully the same distribution as a given dataset. GG-GAN generates points in an euclidian space that is then turned into a graph using a similarity function on the space. This approach is justified by Theorem 1. The authors show that their method successfully generate graphs within the same scope as the input dataset, and show that GG-GAN generates much more new graphs that current state of the art approach.

Pros:
+ The paper is well written, honnest and didactic.
+ The proposed method is new and is experimentaly efficient.

Cons:
- The experiments could be more convincing: small dimension (<=20), classical graph benchmark.
- The introduction of $\phi$ is confusing, and its implications not well justified.

Remarks:
- Proposition 1 should be clarified with assumption within the theorem statement. The "probability of being drawn by probability $\mathcal{D}$" is not well defined. This should be made very rigourous.
- Section 2.1: 'complete representation' is not defined. Wouldn't k>=n-1 be enough?
- Section 3.1: it is not totally clear to the reader why the set $\phi$ is important in GG-GAN. i do not get how it works. How do you learn such parameters? How does it avoid collisions?
- Section 3.3: some recent published papers actually give simple tricks for MPNNs in order to achieve universality without going through higher order tensors [1, 2]. It would be interesting to investigate the relationship of these approachs with the concatenation trick in GG-GAN.
- I am not fully convinced the ethical consideration and impact paragraph is needed here.

Questions:
- Experiments: what are the number of non-isomoprhic classes in the considered datasets?
- You finally sample through a Bernouilli hence non directly differentiable. Why not using the same trick with other methods instead of going for GANs?
- How does $\phi$ looks like after training? What happens if we do not learn it and use random vectors instead?
- When learning several $\phi$ ("a batch") how do you precisely use it for generation?

Typos:
- many references are incomplete, e.g. "A variational inequality perspective on generative adversarial networks" is ICLR'17, "Improved training of Wassertstein GANs" is NeurIPS'17, etc
- ref 'On random graph', Rényi: typo on the accent

[1] Coloring graph neural networks for node disambiguation, IJCAI'20
[2] Universal Invariant and Equivariant Graph Neural Networks, NeurIPS'19

---

### Official Review · AnonReviewer4 · 2020-10-28
**The paper claimed multiple fascinating properties of GG-GAN, but lacks some supporting analysis and experimental demonstration.**

**Rating:** 5
**Confidence:** 4

**Review:**

The paper proposes GG-GAN, a GAN-based graph generative model for mimicking the structure distribution of realistic networks.  The authors claimed multiple fascinating properties of GG-GAN, including (1) isomorphism consistency, (2) expressive power, (3) scalability, (4) novelty. Experimental results on three datasets show the performance of GG-GAN in terms of five network properties. In general, the paper is well written and easy-to-follow. And a bunch of add-on theorems is provided to support the rationality of the proposed method. My major concerns are that the paper might over-claimed their contributions and the experimental results show limited improvement over the baseline methods. Here are the detailed comments:

[Novelty]: The studied problem has been (partially) studied in the existing literature (e.g., isomorphism in graphRNN, expressive power in NetGAN, Scalability in TagGen, etc. ). It is unclear to me what the key contributions of this paper on top of existing work.

[Literature review] The authors fail to provide a comprehensive literature review in the context of graph generation. The authors might want to provide an individual section to fully discuss the connection between this paper to the previous work. For example,
* in terms of isomorphism, what is the difference between GG-GAN and the way employed in graphRNN? Where are the key innovations?
* in terms of scalability, the authors claimed that "GG-GAN is significantly faster than autoregressive models". Have you compared with the recent proposed Transformer-based graph generative model (e.g., TagGen)?
* ...

[Theoretical analysis] The paper includes a bunch of add-on theorems. But, it seems to me that the connection between them and the proposed GG-GAN is weak.

[Experiments - network quality] The paper shows limited improved over the baseline methods in terms of the quality of the generated graph. Interestingly, the paper fails to compare with NetGAN, which achieves SoTA performance in a list of network properties in the real-world datasets in my practice.

[Experiments - scalability] As scalability is one of the major claims of GG-GAN - "GG-GAN is significantly faster than the SoTA autoregressive models", the authors should give a thorough comparison with the existing RNN-based graph generative model (e.g., NetGAN)/Transformer-based graph generative model (e.g., TagGen). Moreover, the authors may want to provide some insightful discussion on why the proposed GAN-based is definitely faster than the autoregressive model. To my best of knowledge, the autoregressive models can be mostly scaled with O(kn), while the GAN-based graph generators are required to fully specify the adjacency matrix A with an O(n^2) parameter space. Please correct me if I am wrong here.


Overall, I enjoyed reading this paper. But, without clearing my concerns stated above, I will vote for weak rejection.

---

### Official Review · AnonReviewer3 · 2020-10-29
**GG-GAN: A GEOMETRIC GRAPH GENERATIVE ADVERSARIAL NETWORK**

**Rating:** 6
**Confidence:** 1

**Review:**

The main contribution of this paper is dubbed as the geometric graph (GG) generative adversarial network (GAN), which is a Wasserstein GAN that addresses the challenges.
The proposed method is inspiring and has sufficient theoretical support. This paper is globally well organized and clearly written.

---

### Official Review · AnonReviewer2 · 2020-10-29
**Interesting problem and a neat idea, but some essential parts are missing.**

**Rating:** 5
**Confidence:** 4

**Review:**

In general, this paper deals with an interesting and essential problem to generate geometric graphs under several standards. The whole algorithm seems easy to implement or reproduce. It seems with minor modifications to traditional autoregressor based generative graph models, the proposed framework can effectively model isomorphism as well as delivers certain novelty. The idea of the paper is with novelty and some theorems can support the observations.

However, I still have several concerns about the paper, stated as follows:

1) While the paper emphasized that the proposed GG-GAN is capable of producing geometric graphs with under several vital criteria (e.g. novelty, scalability and modeling complex dependencies), one critical factor is missing: mode collapse/generation diversity. Many of the generative models still suffer from mode collapse/generation diversity problem, resulting in a small portion of generated variants than empirical observations. I would recommend the authors to discuss and give more evidence showing the ability of the proposed method to avoid such pitfall.

2) The authors claimed that the proposed method can model complex local and global dependencies among nodes and edges. I understand that such a procedure can handle node dependencies given the design of the generator part. However, I suspect the ability of the proposed method to model complex "edge dependencies". To me, the sampling procedure of edges is performed under independent Bernoulli distribution for each edge. Therefore, it's inappropriate the claim that GG-GAN incorporates any mechanism to model dependencies between edges.

3) In corollary 1, the authors proved the existence of some distribution \mathcal{D}_x. However, it seems that existence of such distribution is naive: we can simply establish distbution with Delta-functions for each node. If I understand right, this corollary is not very informative.

4)  Proposition is under the condition that each node is sampled independently. However, such a sampling mechanism can be easily replaced with a sampling procedure following a point process to avoid the coincide of nodes. I suppose that it would be better to briefly discuss the sampling procedure under this setting. Otherwise, it would be too weak Proposition 1 is.

5) Section 2.2.3 Avoiding Collision is obscure to me. The necessity of such a mechanism is not well understood for me. I suggest the authors to give more details or clarify with some theoretical analysis to justify their claim in this section.

6) Though the authors gave some discussion on the hand-crafted features of nodes, I still think the features employed in GG-GAN is ad-hoc. Hand-crafted features can greatly hinder the capacity of deep learning model and thus weaken the contribution of the paper.

7) I see that to construct the initial input to the generator, an initial fixed point configuration and a unique sampled z for each node are concatenated. I suggest the authors to show what will happen if we sample z separately for each node. This may help to understand the necessity of unique sampling, then further show the mechanism behind it.

I might consider to raise the rating if the authors can address my concerns well.

---

### Official Review · AnonReviewer5 · 2020-11-07
**The paper is interesting and well-written but lacks good numerical evidence section**

**Rating:** 5
**Confidence:** 3

**Review:**

The work proposes to use WGAN architecture to learn latent space for generating new graphs with similar properties to the original ones. The authors show that their model is capable to control a probability of each new generated graph. Moreover it’s equivariant function which ensures that isomorphic graphs have the same probability to be generated. These properties are desirable  if we want to generate efficiently new graphs with properties similar to the graphs in the training set.

The paper is interesting and well-written, designs important properties for the generator of graphs. Usage of GAN in graph generator has been explored in the previous approaches and showed good results for generating graphs with similar properties to a given one [1].

My main concern is in numerical evidence section.

•       Datasets. There are essentially 2 real-world datasets with 9 vertices graphs and one artificial dataset with 20 vertices. If a graph has only 9 vertices, there are 12346 non-isomorphic graphs [2]. Your datasets is composed of around 10K graphs, therefore they should have a large portion of isomorphic graphs in train and test. Such repetitions may negatively affect performance of your model as well as baselines [3]. Also, since the graphs are small and consequently the number of non-isomorphic graphs is small, it’s possible to generate all of them quite easily for graph discovery – the main motivation of this paper. It would be more convincing to have a comparison on medium and big graphs.

•       Baselines. Table 1 is not convincing in showing that the proposed method is better than preious approaches. There is a single result (out of 15) where MMD is better than other methods. graphRNN achieves 10 best or second best performances (vs 7 of GG-GAN). Additionally, I would be curious to see other graph generators such as NetGAN [1] (which should scale well for these sizes of graphs) and even simpler baselines such as finding parameters from train set of simple network models (Watts-Strogatz, Barabási–Albert model, Chung-Lu, etc.) and then generating random graphs from these models.

[1] NetGAN: Generating Graphs via Random Walks

[2] http://oeis.org/A000088

[3] Understanding Isomorphism Bias in Graph Data Sets
 https://arxiv.org/abs/1910.12091

---

### Decision · Program_Chairs · 2021-01-07
**Final Decision**

**Decision:**

Reject

**Comment:**

In this paper, the authors proposed a geometric graph generator that applies a WGAN model for efficient geometric interpretation. All the reviewers agree that the idea is interesting and the method has the potentials for graph generation tasks. Unfortunately, the experimental part is unsatisfying, which makes the paper on the borderline. More analytic experiments should be designed to verify the properties of the proposed GG-GAN, especially its scalability. Although in the rebuttal phase the authors add a simple example to generate large but simple graphs, we would like to see more experiments and comparisons on more real-world large graphs (even if the performance may not be good, the results will be constructive for both readers and authors to understand the work).